# Machine Learning Algorithm-Based Prediction Model for the Augmented Use of Clozapine with Electroconvulsive Therapy in Patients with Schizophrenia

**DOI:** 10.3390/jpm12060969

**Published:** 2022-06-14

**Authors:** Hong Seok Oh, Bong Ju Lee, Yu Sang Lee, Ok-Jin Jang, Yukako Nakagami, Toshiya Inada, Takahiro A. Kato, Shigenobu Kanba, Mian-Yoon Chong, Sih-Ku Lin, Tianmei Si, Yu-Tao Xiang, Ajit Avasthi, Sandeep Grover, Roy Abraham Kallivayalil, Pornjira Pariwatcharakul, Kok Yoon Chee, Andi J. Tanra, Golam Rabbani, Afzal Javed, Samudra Kathiarachchi, Win Aung Myint, Tran Van Cuong, Yuxi Wang, Kang Sim, Norman Sartorius, Chay-Hoon Tan, Naotaka Shinfuku, Yong Chon Park, Seon-Cheol Park

**Affiliations:** 1Department of Psychiatry, Konyang University Hospital, Daejeon 35356, Korea; ohssho@kyuh.ac.kr; 2Department of Psychiatry, Inje University Haeundae Paik Hospital, Busan 48108, Korea; bongjulee@empal.com; 3Department of Psychiatry, Yong-In Mental Hospital, Yongin 17089, Korea; yusanglee@gmail.com; 4Department of Psychiatry, Bugok National Hospital, Changyeong 50365, Korea; applegm@hanmail.net; 5Department of Psychiatry, Kyoto University Graduate School of Medicine, Kyoto 606-8501, Japan; nakagami.yukako.4s@kyoto-u.ac.jp; 6Department of Psychiatry, Nagoya University Graduate School of Medicine, Nagoya 466-8550, Japan; toshiya.inada@gmail.com; 7Department of Neuropsychiatry, Graduate School of Medical Sciences, Kyushu University, Fukuoka 812-8582, Japan; takahiro@npsych.med.kyushu-u.ac.jp (T.A.K.); kanba.shigenobu.921@m.kyushu-u.ac.jp (S.K.); 8Department of Psychiatry, Kaohsiung Chang Gung Memorial Hospital, Kaohsiung & Chang Gung University School of Medicine, Taoyuan 83301, Taiwan; mchong@cgmh.org.tw; 9Department of Psychiatry, Linkou Chang Gung Memorial Hospital, Taoyuan 33305, Taiwan; daf68@tpech.gov.tw; 10Peking Institute of Mental Health (PIMH), Peking University, Beijing 100083, China; si.tian-mei@163.com; 11Unit of Psychiatry, Department of Public Health and Medicinal Administration & Institute of Translational Medicine, Faculty of Health Sciences, University of Macau, Macao SAR, China; xyutly@gmail.com; 12Department of Psychiatry, Post Graduate Institute of Medical Education and Research, Chandigarh 160012, India; drajitavasthi@yahoo.co.in (A.A.); drsandeepg2002@gmail.com (S.G.); 13Department of Psychiatry, Pushpagiri Institute of Medical Sciences, Tiruvalla 689101, India; roykalli@gmail.com; 14Department of Psychiatry, Faculty of Medicine Siriraj Hospital, Mahidol University, Bangkok 10400, Thailand; pornjirap@gmail.com; 15Tunku Abdul Rahman Institute of Neuroscience, Kuala Lumpur Hospital, Kuala Lumpur 502586, Malaysia; cheekokyoon@yahoo.com; 16Wahidin Sudirohusodo University, Makassar 90245, Sulawesi Selatan, Indonesia; ajtanra@yahoo.com; 17National Institute of Mental Health, Dhaka 1207, Bangladesh; rabbanigolam33@gmail.com; 18Pakistan Psychiatric Research Centre, Fountain House, Lahore 39020, Pakistan; afzalj@gmail.com; 19Department of Psychiatry, University of Sri Jayewardenepura, Nugegoda 10250, Sri Lanka; samudratk@gmail.com; 20Department of Mental Health, University of Medicine (1), Yangon 15032, Myanmar; dr.winaungmyint@gmail.com; 21National Psychiatry Hospital, Hanoi 10000, Vietnam; hoitamthanhoc@gmail.com; 22West Region, Institute of Mental Health, Singapore 119228, Singapore; yuxi.wang@mohh.com.sg (Y.W.); ksim6133@gmail.com (K.S.); 23Research Division, Institute of Mental Health, Singapore 119228, Singapore; 24Association of the Improvement of Mental Health Programs (AMH), 1209 Geneva, Switzerland; sartorius@normansartorius.com; 25Department of Pharmacology, National University Hospital, Singapore 119228, Singapore; chay_hoon_tan@nuhs.edu.sg; 26Department of Social Welfare, School of Human Sciences, Seinan Gakuin University, Fukuoka 814-8511, Japan; shinfukunaotaka@gmail.com; 27Department of Psychiatry, Hanyang University College of Medicine, Seoul 04763, Korea; hypyc@hanyang.ac.kr; 28Department of Psychiatry, Hanyang University Guri Hospital, Guri 11923, Korea

**Keywords:** schizophrenia, clozapine, electroconvulsive therapy (ECT), augmentation, machine learning, precision medicine

## Abstract

The augmentation of clozapine with electroconvulsive therapy (ECT) has been an optimal treatment option for patients with treatment- or clozapine-resistant schizophrenia. Using data from the Research on Asian Psychotropic Prescription Patterns for Antipsychotics survey, which was the largest international psychiatry research collaboration in Asia, our study aimed to develop a machine learning algorithm-based substantial prediction model for the augmented use of clozapine with ECT in patients with schizophrenia in terms of precision medicine. A random forest model and least absolute shrinkage and selection operator (LASSO) model were used to develop a substantial prediction model for the augmented use of clozapine with ECT. Among the 3744 Asian patients with schizophrenia, those treated with a combination of clozapine and ECT were characterized by significantly greater proportions of females and inpatients, a longer duration of illness, and a greater prevalence of negative symptoms and social or occupational dysfunction than those not treated. In the random forest model, the area under the curve (AUC), which was the most preferred indicator of the prediction model, was 0.774. The overall accuracy was 0.817 (95% confidence interval, 0.793–0.839). Inpatient status was the most important variable in the substantial prediction model, followed by BMI, age, social or occupational dysfunction, persistent symptoms, illness duration > 20 years, and others. Furthermore, the AUC and overall accuracy of the LASSO model were 0.831 and 0.644 (95% CI, 0.615–0.672), respectively. Despite the subtle differences in both AUC and overall accuracy of the random forest model and LASSO model, the important variables were commonly shared by the two models. Using the machine learning algorithm, our findings allow the development of a substantial prediction model for the augmented use of clozapine with ECT in Asian patients with schizophrenia. This substantial prediction model can support further studies to develop a substantial prediction model for the augmented use of clozapine with ECT in patients with schizophrenia in a strict epidemiological context.

## 1. Introduction

Approximately one-third of patients with schizophrenia respond poorly to standard antipsychotic medication treatments [1,2]. Although clozapine has been commonly considered an effective antipsychotic medication option for patients with treatment-resistant schizophrenia (TRS) [3,4], approximately half to three-quarters of these patients are resistant to clozapine treatment [3,5]. In addition, electroconvulsive therapy (ECT) has been an effective treatment option for catatonia, suicidality, aggression, and neuroleptic malignant syndrome [6]. In terms of an augmentation of clozapine with ECT, clozapine may reduce the seizure threshold [7]. The combination of clozapine and ECT is considered one of the effective and safe treatment options for TRS (especially clozapine-resistant schizophrenia [CRS]) [8] according to the following reasons: despite its poor understanding, increased monoamine release; the stimulation of pituitary secretion of thyroid-stimulating hormone, adrenocorticotropic hormone, endorphins, and prolactin; and increased neurogenesis have been proposed as mechanisms for the augmentation of clozapine with ECT [9]. In addition, a systematic review and meta-analysis of open and randomized controlled trials, retrospective chart reviews, case series, and case reports demonstrated that approximately two-thirds (*n* = 83, 66%) of 126 patients with TRS met the response criterion to a combination of clozapine and ECT. In contrast, 14% of patients have shown adverse events (i.e., cognitive dysfunction, postictal confusion, delirium, seizure) [10]. In addition, a randomized single-blind 8-week study of patients with CRS reported that approximately half met the response criterion in the ECT augmentation group, whereas none met the criterion in the clozapine group. No differences between the two groups were reported in terms of side effects and cognitive effects [8]. Furthermore, the observational studies presented that an augmentation of clozapine with ECT can rapidly and substantially improve the psychotic symptoms of patients with TRS [11,12].

Given the potential biomarkers, clinical characteristics, and cognitive features of schizophrenia in terms of neural circuit theory, the four-stage model was proposed by Insel [13]. The characteristics of the four stages are as follows: Stage I is characterized by genetic vulnerability and environmental exposure, which can be diagnosed with genetic sequence and family history, and few cognitive deficits. Stage II is characterized by cognitive and behavioral deterioration, and help seeking; it can be diagnosed with cognitive assessment, neuroimaging, and the Structured Interview for Prodromal Syndrome and can be supported by polysaturated fatty acid consumption and social support. Stage III is characterized by features of abnormal thought and behavior and a relapsing–remitting course, which can be diagnosed through clinical interviews and treated with medications and psychosocial interventions. Stage IV is characterized by continued deterioration, unemployment, homelessness, medical complications, and incarceration, which can be treated with medication, psychosocial interventions, and rehabilitation services. However, in terms of precision medicine in psychosis, to the best of our knowledge, few clinically valid and reliable biomarkers of its onset, course, and outcome have been identified [14]. Although it has been expected that the use of clozapine augmentation with ECT can be optimized in terms of the four-stage model of schizophrenia proposed by Insel [13], its use is still based only on the clinical characteristics and cognitive features of individual patients. In addition, it is necessary to develop a machine learning-based substantial prediction model for the augmentation of clozapine with ECT based on the sociodemographic, clinical, and symptomatic characteristics of patients with schizophrenia.

It is hypothesized that the use of clozapine augmentation with ECT is rare in patients with schizophrenia. In addition, machine learning can provide meaningful insights and applications by offering promising solutions to harness big data [15]. However, to the best of our knowledge, substantial prediction models based on machine learning for the use of a combination of clozapine and ECT have rarely been developed. In terms of precision medicine, using data from the Research on Asian Psychotropic Prescription Pattern for Antipsychotics, phase 4 (REAP-AP4) [16,17,18], which has been the largest international collaborative study in the realm of psychiatry in Asia, we aimed to develop a machine learning-based substantial prediction model for the augmented use of clozapine with ECT in Asian patients with schizophrenia. However, data from the REAP-AP survey have been obtained temporally but not longitudinally. Therefore, we aim to achieve a substantial prediction model.

## 2. Materials and Methods

### 2.1. Study Overview and Participants

As described in previous studies [16,17,18], the REAP-AP4 survey aimed to examine the patterns of psychotropic medication use and their clinical correlates and explore methods to improve the patterns of psychotropic medication use in Asian patients with schizophrenia. A total of 3744 patients with schizophrenia were consecutively recruited from March to June 2016 from 71 survey centers in 15 Asian countries and special administrative regions (SARs), including Bangladesh, China, Hong Kong, India, Indonesia, Japan, Korea, Malaysia, Myanmar, Pakistan, Singapore, Sri Lanka, Taiwan, Thailand, and Vietnam. Study participants were enrolled using convenience sampling. The Institutional Review Board of Taipei City Hospital, Taipei, Taiwan (No. TCHIRB-10412128-E), and other survey hospitals approved the study protocol and informed consent forms. All study participants completed the written informed consent forms before enrolment. A conference meeting to increase the consistency of data collection and confirm the diagnosis of schizophrenia among the survey centers was held prior to the initiation of the study. As per the protocol, detailed sociodemographic, clinical, symptomatic, and psychiatric treatment-related characteristics were collected by the study coordinators trained and supervised by clinical psychiatrists, at the survey centers. Data collected from the study participants were stored on the REAP survey website.

To examine psychotropic medication use patterns in Asian patients with schizophrenia in the real world, the inclusion and exclusion criteria were defined. The inclusion criteria were as follows: (i) diagnosis of schizophrenia, based on the Diagnostic and Statistical Manual of Mental Disorders, fifth edition (DSM-5) [19], confirmed by clinical psychiatrists at the survey centers; (ii) treatment with antipsychotic medications (N05A), defined by the Anatomical Therapeutic Chemical classification index [20]; and (iii) availability of information on current clozapine medication and ECT application. The exclusion criteria were as follows: (i) comorbidity of severe physical disease and (ii) inability to read or write.

### 2.2. Variable Profiles for the Substantial Prediction Model

In the study, we used sociodemographic data (i.e., age, sex, body mass index [BMI]), clinical data (i.e., outpatient or inpatient status, duration of illness [<3 months, 3–6 months, 6–12 months, 1–5 years, 5–10 years, 10–20 years, or >20 years], clinical course for the past 1 year [remission or persistent symptoms]), and current psychopathological symptoms (i.e., delusion, hallucination, disorganized speech, grossly disorganized or catatonic behavior, negative symptoms, social or occupational dysfunction, verbal aggression, physical aggression, significant affective symptoms). A predictive utility of BMI for metabolic syndrome was approved among Japanese patients with schizophrenia [21]. BMI, which might be inversely associated with homocysteine level, was proposed to indicate clinical symptoms and glucose and lipid levels among Chinese patients with schizophrenia [22]. BMI was positively associated with positive symptoms among antipsychotic-naïve schizophrenia patients [23]. In addition, a differential association between BMI and fronto-limbic white matter microstructure was found among patients with first-episode schizophrenia spectrum disorders [24]. Furthermore, BMI and age had the moderating effects of an association between a history of suicidal attempts and COVID-19 infection among patients with schizophrenia or schizoaffective disorder [25]. Therefore, BMI and other sociodemographic and clinical data were included as one of the variable profiles for the substantial prediction model. The current symptoms were defined based on the DSM-5 [19] and evaluated using a dichotomous value manner (presence or absence).

### 2.3. Data Processing and Machine Learning

All patients with missing data were excluded from the analysis. Furthermore, all data were divided into training (0.7) and test (0.3) sets in a 7:3 ratio of 3744 patients. A 10-fold cross-validation was used to train the entire training dataset within the training set. Because the augmentation of clozapine with ECT is rarely used, the synthetic minority oversampling technique was used [26]. In addition, the area under the curve (AUC) of the receiver operating characteristic curve was used to determine the main performance. Moreover, the overall accuracy, sensitivity, specificity, negative predictive value, and positive predictive value were calculated. The classification and regression training package of R was used to perform hyperparameter tuning, confusion matrix composition, and AUC assessment [27,28]. The random forest model and least absolute shrinkage and selection operator (LASSO) model were used to compare substantial prediction models for a combination of clozapine and ECT among Asian patients with schizophrenia. 

First, a random forest model was used as the algorithm in this analysis. Although the decision tree model can indicate variable importance and classification mechanisms in an intuitive manner, and its computing cost is low, it is vulnerable to overfitting. Thus, the decision tree model is rarely used as an algorithm. Random forest is the ensemble model used to overcome the limitations of the decision tree model [29]. Its predictive power is maximized under the condition that new data (i.e., test set) are provided rather than used for training by minimizing the overfitting of the decision tree. According to the principle of “mean decreases in accuracy,” the variable importance was calculated to identify the variables that significantly contributed to the optimization of the predictive model [30]. The importance of variable *j* was estimated as follows: Several cases were not sampled and were called out-of-bag because bagging permitted cases to be sampled more than once for the same classifier. After handing down the out-of-bag samples to the tree, the prediction accuracy decreased with training using one of the trees. Subsequently, the change in variable *j* was conducted in the out-of-bag samples of trees, and the accuracy was calculated again. Before and after the permutation over all trees, the raw score for the importance of variable *j* was calculated by computing the average gap in out-of-bag errors. The return of the score to normal was calculated using the standardized deviation of the difference. Finally, the score was reduced so that the minimum and maximum values were set to 0 and 100, respectively. Using care, the attainment of a confusion matrix, hyperparameter tuning, and performance measurements were conducted. The prediction performance of the out-of-bag portion of the data was recorded for each tree. The measurement and recording of the prediction performance were conducted each time, while alternative permutations of the predictors were conducted. Based on all the trees, the average value of the difference between these two prediction performances was obtained and normalized. The final prediction performance was measured using test data, which were not used in the training phase.

Second, the LASSO model was also used in this analysis. The LASSO model used a regularization term *λE(**ω) =*
*λΣ |ω_k_|* [31]. The LASSO model can be practically used as a feature reduction method, as the coefficients of weak predictive variables decrease to zero. In this analysis, the hyperparameter, which inversely reflected the strength of the regularization parameter *λ*, was set to 0.0076. In addition, the penalty option was set to “l1,” and other hyperparameters were set to default in the logistic regression scikit-learn library.

## 3. Results

### 3.1. General Characteristics of the Study Participants

Among the 3744 Asian patients with schizophrenia, 1.3% (*n* = 47) were treated with a combination of clozapine and ECT. As shown in Table 1, the patients who were treated with an augmentation of clozapine with ECT were characterized by significantly greater proportions of females (*χ*^2^ = 5.142, *p* = 0.023) and inpatients (*χ*^2^ = 39.942, *p* < 0.0001), a longer duration of illness (*χ*^2^ = 19.253, *p* = 0.004), and a greater prevalence of negative symptoms (*χ*^2^ = 8.571, *p* = 0.003) and social or occupational dysfunction (*χ*^2^ = 3.960, *p* = 0.047) than those treated without clozapine augmentation with ECT.

### 3.2. Substantial Prediction Model Performance and Variable Importance: Random Forest Model

The substantial prediction model was developed for 47 patients treated with a combination of clozapine and ECT among the 3744 patients in this analysis. As shown in Figure 1, the AUC, which was the most preferred indicator of the prediction model, was 0.774 in the random forest model to predict the use of a combination of clozapine and ECT. The overall accuracy, sensitivity, specificity, positive predictive value, and negative predictive value were 0.817 (95% confidence interval [CI], 0.793–0.839), 0.049, 0.991, 0.821, and 0.556, respectively. As shown in Figure 2, inpatient status was the most important variable in the substantial prediction model, followed by BMI, age, social or occupational dysfunction, persistent symptoms, illness duration > 20 years, and others.

### 3.3. Substantial Prediction Model Performance and Variable Importance: LASSO Model

As shown in Figure 3, the AUC was 0.831 in the LASSO model to predict the use of a combination of clozapine and ECT. The overall accuracy, sensitivity, specificity, positive predictive value, and negative predictive value were 0.644 (95% CI, 0.615–0.672), 0.033, 0.993, 0.722, and 0.643, respectively. As shown in Figure 4, inpatient status was the most important variable in the substantial prediction model, followed by significant affective symptoms, illness duration > 20 years, disorganized speech, persistent symptoms, social or occupational dysfunction, and others.

## 4. Discussion

A substantial prediction model has been developed for the augmented use of clozapine with ECT with 47 of the 3744 Asian patients with schizophrenia using machine learning algorithms of both the random forest and LASSO models. In the random forest model, most importantly, the finding that inpatient status is the most important variable in the prediction model has several clinical implications. First, it is hypothesized that recruitment as “inpatient” may be associated with a “moderately ill” or “markedly ill” severity level of schizophrenia because of the following reasons: in a prospective and randomized study of participants administered ECT plus clozapine (*n* = 20) [7], the mean total and psychotic symptom subscale scores on the Brief Psychiatric Rating Scale were 45.7 (standard deviation [SD] = 1.9) and 16.6 (SD = 0.9), respectively, which indicate “moderately ill” severity according to the framework by Leucht et al. [32]. In addition, in study participants receiving the combination of clozapine and ECT (*n* = 14), the mean total score on the Positive and Negative Syndrome Scale was 101.7 (SD = 13.9), which indicates a “markedly ill” severity based on the implication by Leucht et al. [33]. Moreover, negative symptoms and social or occupational dysfunction were significantly more prevalent in the group with the augmented use of clozapine with ECT than in the group without. Second, the “inpatient” status may be associated with TRS because of the following reasons: according to the definition of the Treatment Response and Resistance in Psychosis group [34,35], TRS denotes the nonresponse to a trial of two antipsychotics for 6 or more weeks each at a dose equivalent to more than 600 mg of chlorpromazine, which is the daily dose among patients with schizophrenia. TRS is clinically characterized by a longer duration of illness, more severe negative symptoms, poor premorbid social adjustment, and lack of illness insight [36,37,38], which are consistent with the findings. In addition, clozapine is considered the most effective pharmacological treatment option for TRS [39]. Third, the “inpatient” status may be associated with CRS because of the following reasons: CRS denotes the nonresponse to 200–300 mg of clozapine daily with blood levels > 350 ng/mL for 2–3 months among patients with schizophrenia [40,41,42,43]. CRS is considered the most severely ill condition of schizophrenia and is clinically characterized by a greater severity of positive and negative symptoms and lower quality of life [44,45]. Common treatments for CRS include antipsychotics [45], mood stabilizers or antidepressants [46,47,48,49,50], and ECT [3,4,5,6,7,8,9,10,11,12] augmentations. Herein, the “inpatient” status may be associated with a “moderately ill” or “markedly ill” severity level of TRS or CRS, which corresponds to stage IV of Insel’s four-stage model [13]. In the random forest model, other predictive variables for the augmented use of clozapine were BMI, age, social or occupational dysfunction, persistent symptoms, and duration of illness > 20 years. The BMI was mainly affected by clozapine-induced weight gain. A recent meta-analysis showed that clozapine can result in weight gain of approximately 3 kg after a median duration of 6 (range, 2–13) weeks [51]. Weight gain is clinically associated with lower quality of life, increased morbidity and mortality, and patient nonadherence [52]. In several studies on the efficacy of clozapine augmentation with ECT, the mean duration of clozapine treatment ranged from 3.9 (SD = 3.4) to 6.4 (SD = 6.1) years [11,12,48,49]. In addition, age, social or occupational dysfunction, persistent symptoms, and illness duration >20 years were all associated with the clinical characteristics of TRS or CRS. As previously mentioned, social or occupational dysfunction and persistent symptoms can be associated with the clinical characteristics of TRS or CRS [28,29,34,35,36,37]. In addition, several studies assessing the efficacy of the augmented use of clozapine with ECT presented that CRS patients’ mean duration of illness ranged from 15.0 (SD = 8.6) to 22.8 (SD = 14.2) years [11,12,53,54]. In patients with TRS or CRS, age may be affected by a longer duration of illness and longer duration of untreated psychosis, and social or occupational dysfunction and persistent symptoms may be affected by clinical characteristics.

In the random forest model, regarding the performance of the substantial prediction model for the augmented use of clozapine with ECT, the AUC and accuracy were 0.774 and 0.817, respectively. In general, AUCs of 0.5, 0.7–0.8, 0.8–0.9, and >0.9 are considered no discrimination, acceptable, excellent, and outstanding, respectively [55]. Thus, the performance of our substantial prediction model is considered acceptable. In addition, an AUC of 0.714 corresponds to Cohen’s *d* of 0.800 [56]. However, the low sensitivity value needs to be overcome in future studies. Since the augmentation of clozapine with ECT is rarely used in clinical psychiatric practice, to the best of our knowledge, its rate has rarely been reported in previous epidemiological studies. Thus, although the sensitivity value is insufficient as a par value, using a machine learning-based algorithm, our findings allow the reporting of the specificity value for the substantial prediction model of the augmented use of clozapine with ECT.

In the LASSO model, inpatient status is also the most important variable for the substantial prediction model for using a combination of clozapine and ECT, significant affective symptoms, illness duration > 20 years, disorganized speech, persistent symptoms, social or occupational dysfunction, and others. Although subtle differences presented themselves in the rank orders, the important variable profiles for the LASSO regression-based model tended to be similar to those for the random forest model. In the LASSO model, significant affective symptoms were the second most important variable for the substantial prediction model. Greater affective components can be one of the symptoms in less rigorously defined TRS [57]. In terms of the prediction of the substantial prediction model, the AUC (0.831) is considered an excellent level [55]. Although the AUC of the LASSO model is subtly higher than that of the random forest model, the accuracy of the LASSO model (0.644) is less than that of the random forest model (0.817). Despite the subtle differences in both the AUC and overall accuracy of the random forest model and LASSO model, the important variables were commonly shared by the two models.

Our study has some limitations. First, the study participants of the REAP-AP4 survey were not recruited using a convenience sampling method but rather using a strict epidemiological context. We found a trend for psychotropic drug use patterns, including the augmented use of clozapine with ECT in many Asian countries. However, the significance of the findings from the REAP-AP4 survey might be limited in a strict epidemiological context. Second, current symptoms (i.e., delusion, hallucination, disorganized speech, grossly disorganized or catatonic behavior) were evaluated using a dichotomous value, not a continuous value. Thus, the importance of current symptoms in the substantial prediction model may be affected by their manner of evaluation. Third, we cannot exclude the possibility that our findings were affected by a selection bias. The rate of the augmented use of clozapine with ECT has significantly differed across the 10 Asian countries or SARs in our study. In addition, although the REAP survey is the largest international collaborative psychiatry study in Asia, it was conducted mainly in university-affiliated or training hospitals. Thus, actual clinical psychiatric situations in Asia may be limited to the findings of the REAP-AP survey. Fourth, study participants’ previous treatments with some other medication or ECT have been included as variables for the substantial prediction model. Thus, the generalizability of the substantial prediction model may be limited. Fifth, the duration or treatment of the augmented use of clozapine with ECT was not evaluated in our study. Thus, the substantial prediction model of the treatment outcome of the augmented use of clozapine with ECT was not developed in our study. Sixth, the operational definition of treatment resistance or clozapine resistance was not used in our study. On the contrary, the augmented use of clozapine with ECT was alternatively used. To overcome these limitations, further studies are required to use structured assessment tools to examine psychotropic drug use patterns and current symptoms in Asia in a strict epidemiological context.

## 5. Conclusions

Despite the limitations of the present study, using the machine learning algorithm, our findings allow for the development of a substantial prediction model for the augmented use of clozapine with ECT in Asian patients with schizophrenia. The substantial prediction model may be the basis of the clinical applicability for indicating the augmented use of clozapine with ECT in terms of precision medicine, although augmentation has rarely been used in real clinical practice. Although the model needs to be improved for sensitivity, in the context of precision medicine for patients with TRS or CRS, the substantial prediction model can be used to elaborate psychopharmacological treatment options. This substantial prediction model can be a basis for further studies to develop the substantial prediction model for the augmented use of clozapine with ECT in patients with schizophrenia.

## Figures and Tables

**Figure 1 jpm-12-00969-f001:**
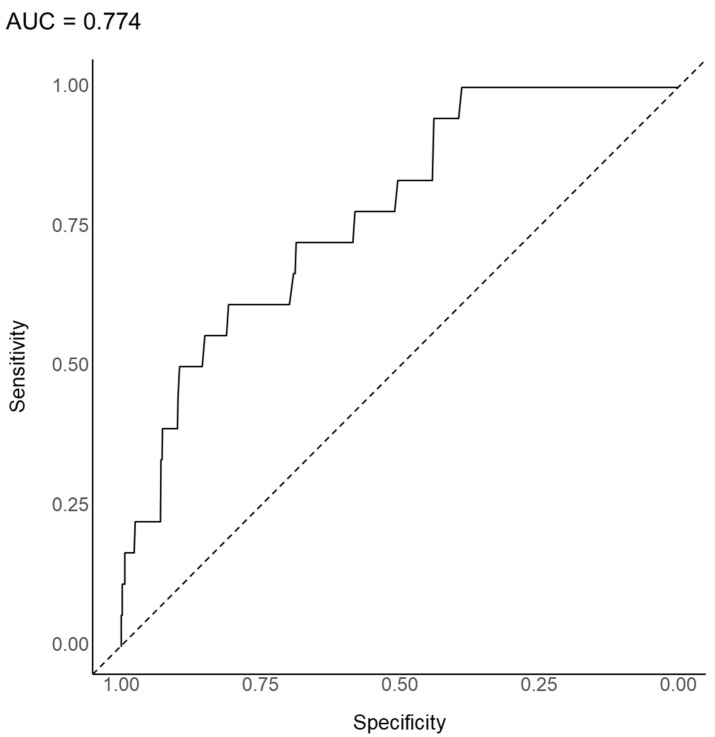
Area under the curve of the receiver operating characteristic curve for the augmented use of clozapine with electroconvulsive therapy in 3744 patients with schizophrenia (random forest model).

**Figure 2 jpm-12-00969-f002:**
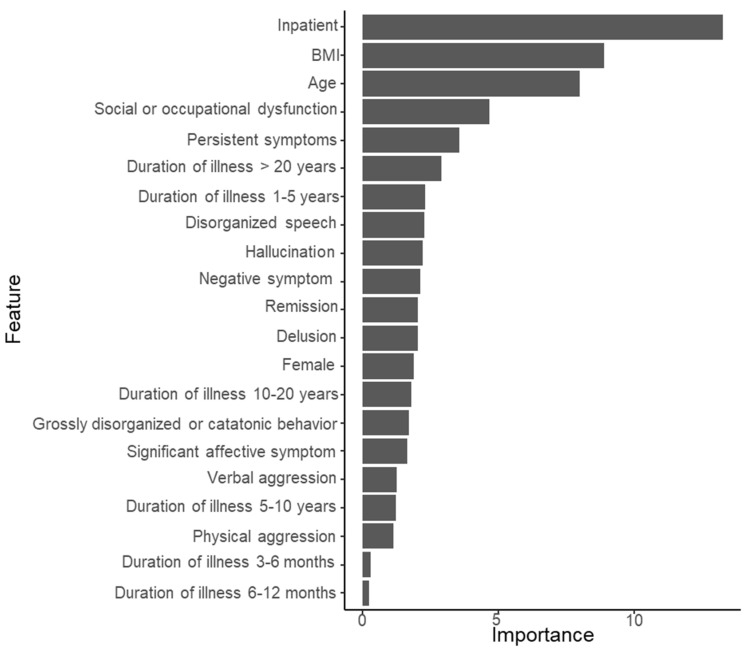
The importance of variables in the prediction model for the augmented use of clozapine with electroconvulsive therapy in 3744 patients with schizophrenia (random forest model). BMI, body mass index.

**Figure 3 jpm-12-00969-f003:**
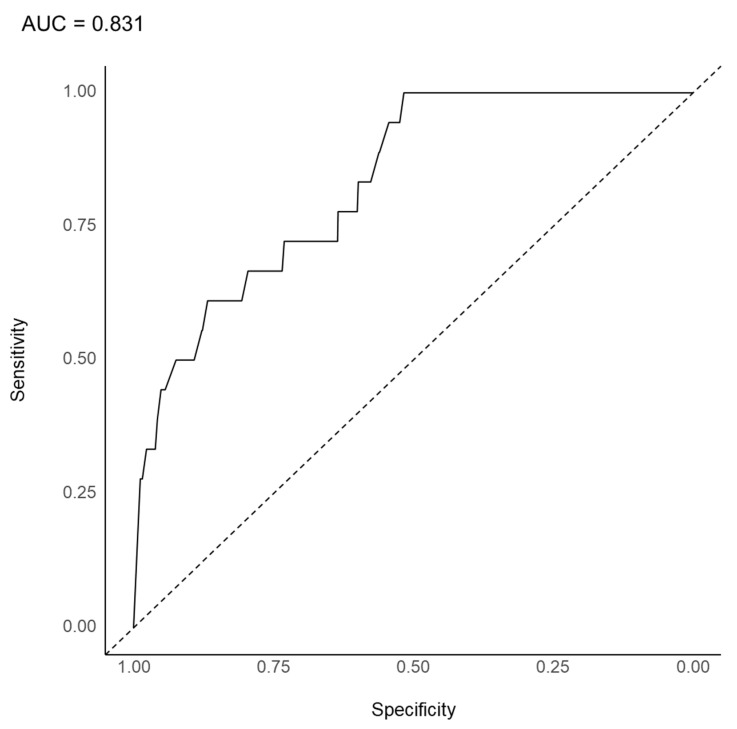
Area under the curve of the receiver operating characteristic curve for the augmented use of clozapine with electroconvulsive therapy in 3744 patients with schizophrenia (LASSO model). LASSO, least absolute shrinkage and selection operator.

**Figure 4 jpm-12-00969-f004:**
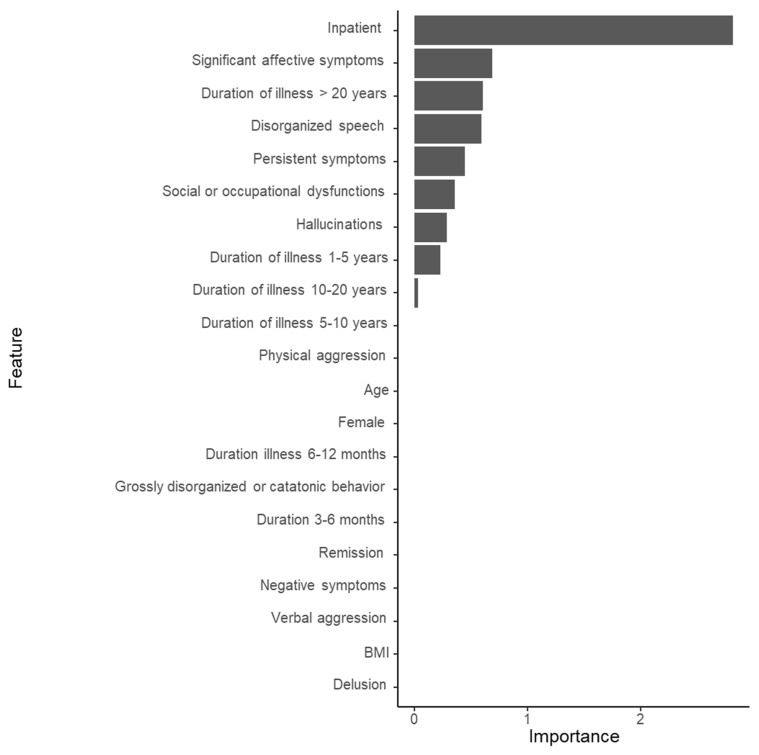
The importance of variables in the prediction model for the augmented use of clozapine with electroconvulsive therapy in 3744 patients with schizophrenia (LASSO model). BMI, body mass index.

**Table 1 jpm-12-00969-t001:** General description of the study participants (*n* = 3744).

	Total (*n* = 3744)	Augmentation of Clozapine with ECT		Statistical Coefficient	*p*-Value
		Yes (*n* = 47)	No (*n* = 3697)		
Country/SAR				χ^2^ = 19.616	<0.0001
Bangladesh, n (%)	99 (2.6)	0 (0.0)	99 (2.7)		
China, n (%)	160 (4.3)	20 (42.6)	140 (87.5)		
Hong Kong, n (%)	31 (0.8)	0 (0.0)	31 (0.8)		
India, n (%)	479 (12.8)	2 (4.3)	477 (12.9)		
Indonesia, n (%)	581 (15.5)	10 (21.3)	571 (15.4)		
Japan, n (%)	229 (6.1)	2 (4.3)	227 (6.1)		
Korea, n (%)	131 (3.5)	0 (0.0)	131 (3.5)		
Malaysia, n (%)	305 (8.1)	4 (8.5)	301 (8.1)		
Myanmar, n (%)	164 (4.4)	0 (0.0)	164 (4.4)		
Pakistan, n (%)	298 (8.0)	0 (0.0)	298 (8.0)		
Singapore, n (%)	171 (4.6)	2 (4.3)	169 (4.6)		
Sri Lanka, n (%)	97 (2.6)	1 (2.1)	96 (2.6)		
Thailand, n (%)	322 (8.6)	2 (4.3)	320 (8.7)		
Taiwan, n (%)	403 (10.8)	2 (4.3)	401 (10.8)		
Vietnam, n (%)	274 (7.3)	2 (4.3)	272 (7.4)		
Age, mean (SD) years	39.5 (13.2)	39.3 (13.6)	39.5 (13.1)	t = −0.109	0.913
Sex				χ^2^ = 5.142	0.023
Male, n (%)	2199 (58.7)	20 (42.6)	2179 (58.9)		
Female, n (%)	1545 (41.3)	27 (57.4)	1518 (41.1)		
BMI, mean (SD) kg/m^2^	23.9 (4.7)	23.2 (3.4)	23.9 (4.7)	t = −1.376	0.319
Hospitalization				χ^2^ = 39.942	<0.0001
Outpatient, n (%)	1793 (47.9)	1 (2.1)	1792 (48.5)		
Inpatient, n (%)	1951 (52.1)	46 (97.9)	1905 (51.5)		
Duration of illness				χ^2^ = 19.253	0.004
<3 months, n (%)	161 (4.3)	6 (12.8)	155 (4.2)		
3–6 months, n (%)	125 (3.3)	0 (0.0)	125 (3.3)		
6–12 months, n (%)	199 (5.3)	1 (2.1)	198 (5.4)		
1–5 years, n (%)	794 (21.2)	5 (0.6)	789 (21.3)		
5–10 years, n (%)	729 (19.5)	8 (17.0)	721 (19.5)		
10–20 years, n (%)	971 (25.9)	10 (21.3)	961 (26.0)		
>20 years, n (%)	765 (20.4)	17 (36.2)	748 (20.2)		
Clinical course for the past 1 year					
Remission, n (%)	1262 (33.7)	18 (38.3)	1244 (33.6)	χ^2^ = 0.449	0.503
Persistent symptoms, n (%)	1917 (51.2)	20 (42.6)	1897 (51.3)	χ^2^ = 1.423	0.233
Current symptoms					
Delusion, n (%)	1599 (42.7)	19 (40.4)	1580 (42.7)	χ^2^ = 0.101	0.750
Hallucination, n (%)	1752 (46.8)	19 (40.4)	1733 (46.9)	χ^2^ = 0.776	0.378
Disorganized speech, n (%)	1110 (29.6)	12 (25.5)	1098 (29.7)	χ^2^ = 0.387	0.534
Grossly disorganized or catatonic behavior, n (%)	666 (17.8)	7 (14.9)	659 (17.8)	χ^2^ = 0.273	0.601
Negative symptom, n (%)	1313 (35.1)	26 (55.3)	1287 (34.8)	χ^2^ = 8.571	0.003
Social or occupational dysfunction, n (%)	1693 (45.2)	28 (59.6)	1665 (45.0)	χ^2^ = 3.960	0.047
Verbal aggression, n (%)	942 (25.2)	9 (19.1)	933 (25.2)	χ^2^ = 0.913	0.339
Physical aggression, n (%)	780 (20.8)	11 (23.4)	769 (20.8)	χ^2^ = 0.191	0.662
Significant affective symptoms, n (%)	425 (11.4)	3 (6.4)	422 (11.4)	χ^2^ = 1.168	0.280

BMI, body mass index; ECT, electroconvulsive therapy; SAR, special administrative region; SD, standard deviation.

## Data Availability

Data sharing not applicable.

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
