# Peer review of "Machine Learning Algorithm-Based Prediction Model for the Augmented Use of Clozapine with Electroconvulsive Therapy in Patients with Schizophrenia"

_jpm, 2022, doi:10.3390/jpm12060969_

Round 1
Reviewer 1 Report
In this paper the authors present a study that aimed to develop a machine learning algorithm-based prediction model for the augmented use of clozapine with electroconvulsive therapy in Asian patients with schizophrenia
The subject of the article is relevant and worthy of discussion.
The authors describe the characteristics of the dataset and describe the characteristics of the participants. They describe the variables used in the prediction model. They describe the prediction model performance and variable importance. Some limitations of the study are also presented. The results obtained are interesting although there are some limitations.
However, there are some aspects that should be clarified and/or analyzed with more detail.
The authors give a brief explanation of why they chose to use the random forest model. They just compare with using the decision tree model. However, the study would be enriched if the authors considered other algorithms and compared the results obtained. A quick search allows you to identify several machine learning algorithms that have been used successfully in clinical studies. This section should be covered in more detail.
The dataset includes data from 3744 participants. However, only 47 were treated with a combination of clozapine and ECT. The authors should explain in more detail which participants were included in the training and test sets. 70% and 30% of the 3744? or 70% and 30% of the 47? Were the results presented in figures 2 and 3 obtained based on the set of 3744 participants?
References are appropriate, in number. However, there are some parts of the text that are supported by outdated references. For example: (Page 21 - Introduction) The first two sentences of the Introduction section.
Author Response
- We completely agree with your comments. We have additionally developed the LASSO-based prediction model.
We have added the following sentences in abstract:
A random forest model and least absolute shrinkage and selection operator (LASSO) model were used to develop a substantial prediction model for the augmented use of clozapine with ECT.
Furthermore, the AUC and overall accuracy of the LASSO model were 0.831 and 0.644 (95% CI, 0.615-0.672), respectively. Despite the subtle differences in both AUC and overall accuracy of the random forest model and LASSO model, the important variables were commonly shared by the two models.
We have added the following sentences in materials and methods:
The random forest model and least absolute shrinkage and selection operator (LASSO) model were used to compare substantial prediction models for a combination of clozapine and ECT among Asian patients with schizophrenia:
Second, the LASSO model was also used in this analysis. The LASSO model used a regularization term λE(ω) = λΣ |ωk| [31]. The LASSO model can be practically used as a feature reduction method, as the coefficients of weak predictive variables decrease to zero. In this analysis, the hyperparameter, which inversely reflected the strength of the regularization parameter λ, was set to 0.0076. In addition, the penalty option was set to “l1,” and other hyperparameters were set to default in the logistic regression scikit-learn library.
We have added the following sentences in results:
- Substantial Prediction Model Performance and Variable Importance: LASSO Model
As shown in Figure 3, the AUC was 0.831 in the LASSO model to predict the use of a combination of clozapine and ECT. The overall accuracy, sensitivity, specificity, positive predictive value, and negative predictive value were 0.644 (95% CI, 0.615-0.672), 0.033, 0.993, 0.722, and 0.643, respectively. As shown in Figure 4, inpatient status was the most important variable in the substantial prediction model, followed by significant affective symptoms, illness duration > 20 years, disorganized speech, persistent symptoms, social or occupational dysfunction, and others.
We have added the following sentences in discussion:
In the LASSO model, inpatient status is also the most important variable for the substantial prediction model for using a combination of clozapine and ECT, significant affective symptoms, illness duration > 20 years, disorganized speech, persistent symptoms, social or occupational dysfunction, and others. Although subtle differences have been presented in the rank orders, the important variable profiles for the LASSO regression-based model have tended to be similar to those for the random forest model. In the LASSO model, significant affective symptoms have been the second most important variable for the substantial prediction model. Greater affective components can be one of the symptoms in less rigorously defined TRS [57]. In terms of the prediction of the substantial prediction model, the AUC (0.831) is considered an excellent level [55]. Whereas the AUC of the LASSO model is subtly higher than that of the random forest model, the accuracy of the LASSO model (0.644) is less than that of the random forest model (0.817). Despite the subtle differences in both AUC and overall accuracy of the random forest model and LASSO model, the important variables were commonly shared by the two models.
We have added the figures as follows;
Figure 3. Area under the curve of the receiver operating characteristic curve for the augmented use of clozapine with electroconvulsive therapy in 3,744 patients with schizophrenia (LASSO model)
Figure 4. The importance of variables in the prediction model for the augmented use of clozapine with electroconvulsive therapy in 3,744 patients with schizophrenia (LASSO model)
- We have added the detailed descriptions in results as follows:
Furthermore, all data were divided into training (0.7) and test (0.3) sets in a 7:3 ratio of 3,744 patients.
- We have revised the first two references as follows:
- Joo, S.W.; Kim, H.; Jo, Y.T.; Ahn, S.; Choi, Y.J.; Choi, W.; Park, S.; Lee J. Comparative effectiveness of antipsychotic monotherapy and polypharmacy in schizophrenia patients with clozapine treatment: A nationwide, health insurance data-based study. Eur. Neuropsychopharmacol. 2022 May 9;59:36-44. doi: 10.1016/j.euroneuro.2022.03.010.
- Kanahara, N.; Nakamura, M.; Shiko, Y.;, Kawasaki, Y.; Iyo, M. Are serum oxytocin concentrations lower in patients with treatment-resistant schizophrenia?: A 5-year longitudinal study. Asian J. Psychiatr. 2022 May 6:103157. doi: 10.1016/j.ajp.2022.103157.
Reviewer 2 Report
dear colleagues
thank you very much for your paper
the introduction did not address the problem very adequately it was not clear for me what is the novelty of this study and the aims remain ambiguous with no clear hypothesis if its hypothesized that the use of clozapine augmentation with ECT is rare in patients with schizophrenia this makes the use of ML/random-forest regression of little use and should have use knn approach because
ML methods for detecting rare events
in temporal data is always challenge. please clarify the introduction. see my comment also down about methods.
be aware that many reviews showed that combined of ECT and clozapine in patients with resistant symptoms of psychosis resulted in a rapid and substantial reduction of psychotic symptoms. so make sure the novelty of the study is not only the use of ringing words like ML.
the methodology need to be described better in the paper with little amount of information given to references outside to previous work this is very burdensome for reviewers and readers and should be addressed in the single document presented in summary
section 2.2 listed the variables but did not provide reasoning for each e.g. why bmi was included? is it dose response dependent ?? kindly during revision each variable to be provided with better explanation.
table 1 clearly provide conclusion that this study is based on 47 pts only and I suggest that the entire discussion to be toned down substantially. the first line of discussion is misleading this study was based on 3,744 Asian patients with schizophrenia whom only 47 (1.3%) used combined therapy czp+ECT.
please provide the r script codes as supplumental so reviewers and readers can examine it.
the charts appear to be no r statistical computing production please verify.
Author Response
- We completely agree with your comments. We have revised the manuscript to reveal the novelty of the findings as follows:
We have added the following sentence in introduction:
However, to our best knowledge, substantial prediction models based on machine learning for the use of a combination of clozapine and ECT have rarely been developed.
- We have revised the references as follows:
- Grzenda, A.; Kragulijac, N.V.; McDonald, W.M.; Nemeroff, C.; Torous, J.T.; Alpert, J.E.; Rodgriuez, C.I.; Widge, A.S. Evaluating the machine learning literature: A primer and user’s guide for psychiatrists. Am. J. Psychiatry 2021, 178, 715-729.
- Sicotte, X.B. Lasso Regression: Implementation of Coordinate Descent. Data Science, Machine Learning and Statistics, Implemented in Python. 2018. Available online: https://xavierbourretsicotte.github.io/lasso_implementation.html (accessed on 3 June 2022).
- We have added the following sentences to explain the inclusion of BMI as the variable for the prediction model in methods:
A predictive utility of BMI for metabolic syndrome was approved among Japanese patients with schizophrenia [21]. BMI, which might be inversely associated with homocysteine level, was proposed to indicate clinical symptoms and glucose and lipid levels among Chinese patients with schizophrenia [22]. BMI was positively associated with positive symptoms among antipsychotic-naïve schizophrenia patients [23]. In addition, a differential association between BMI and fronto-limbic white matter microstructure among patients with first-episode schizophrenia spectrum disorders [24]. Furthermore, BMI and age had the moderating effects of an association between a history of suicidal attempts and COVID-19 infection among patients with schizophrenia or schizoaffective disorder [25]. Therefore, BMI and other sociodemographic and clinical data were included as one of the variable profiles for the substantial prediction model.
- We have revised the first sentence in discussion as follows:
A substantial prediction model has been developed for the augmented use of clozapine with ECT with 47 among the 3,744 Asian patients with schizophrenia using machine learning algorithms of both the random forest and LASSO models.
- We have added the R codes for the machine learning algorithms in supplementary materials as follows:
- Supplementary Material
## ML
library(caret);library(ggplot2)
## Read data
data <- readRDS("datafile.RDS")
## Variable
## Dependent: input$dep_ml
## Independent variables: input$indep_ml
## Scaling
vars.factor <- c('d_clozapine_&_ECT', 'inpatient', 'sex', 'duration_from_onset', 'remission', 'persistent symptoms', 'delusions', 'hallucinations', 'disorganized_speech', 'grossly_disorganized_or_catatonic_behavior', 'negative_symptoms', 'social_or_occupational_dysfunctions', 'verbal_aggression', 'physical_aggression', 'significant_affective_symptoms')
for (v in setdiff(names(data), vars.factor)){
data[[v]] <- scale(data[[v]])
}
## training/test set: SMOTE
set.seed(1)
rn.train <- sample(1:nrow(data), size = round(0.7 *nrow(data)), replace = F)
data.train <- DMwR::SMOTE(as.formula(paste(input$dep_ml, "~ .")), data = data[rn.train])
data.test <- data[-rn.train]
## ML method
myControl <- trainControl(
method = "cv", number = 10,
summaryFunction = twoClassSummary,
classProbs = TRUE
)
## If LASSO
tunegrid <- expand.grid(alpha = 1, lambda = 10^seq(-5, 3, length = 200))
rf1 <- train(as.formula(paste0(input$dep_ml, " ~", paste(input$indep_ml, collapse = "+"))), data = data.train, method = "glmnet",
trControl = myControl, tuneGrid = tunegrid)
## If Random forest
tunegrid <- NULL
rf1 <- train(as.formula(paste0(input$dep_ml, " ~", paste(input$indep_ml, collapse = "+"))), data = data.train, method = "rf",
trControl = myControl, tuneGrid = tunegrid)
## Res
pred <- data.frame("Obs" = data.test[[input$dep_ml]], "Pred" = predict(rf1, data.test, type = "prob")[, 2])
obj.ml <- list(obj = rf1, pred = pred, cmat = confusionMatrix(pred$Obs, predict(rf1, data.test), positive = levels(pred$Obs)[2]))
## ROC
obj.roc <- pROC::roc(Obs ~ as.numeric(Pred), data = obj.ml$pred)
pROC::ggroc(obj.roc) + see::theme_modern() + geom_abline(slope = 1, intercept = 1, lty = 2) +
xlab("Specificity") + ylab("Sensitivity") + ggtitle(paste("AUC =", round(obj.roc$auc, 3)))
## Variable Importance
ggplot(varImp(obj.ml$obj, scale = F), width = 0.05) + ggpubr::theme_classic2()
Round 2
Reviewer 2 Report
thank you for addressing my concerns